# Quality of Life and Psychosocial Well-Being among Intersex-Identifying Individuals in Puerto Rico: An Exploratory Study

**DOI:** 10.3390/ijerph20042899

**Published:** 2023-02-07

**Authors:** Caleb Esteban, Derek Israel Ortiz-Rodz, Yesibelle I. Muñiz-Pérez, Luis Ramírez-Vega, Coral Jiménez-Ricaurte, Edna Mattei-Torres, Victoria Finkel-Aguilar

**Affiliations:** 1Ponce Campus, School of Behavioral and Brain Sciences, Ponce Health Sciences University, Ponce, PR 00716, USA; 2Río Piedras Campus, Department of Psychology, University of Puerto Rico, San Juan, PR 00931, USA

**Keywords:** intersexuality, differences in sex development, quality of life, psychological well-being, social well-being

## Abstract

Purpose: Intersex is an umbrella term used to describe the diversity or differences in the characteristics of physical sexual development. Approximately 1.7% of the population are born intersex, and 1 in every 2000 babies at birth presents genital variation. Unfortunately, there is a lack of research on the health of intersex-identifying persons in Latin America. This study aimed to document experiences of discrimination and violence among self-identifying intersex individuals in Puerto Rico and to determine if there is a significant difference in the quality of life, psychological well-being, and social well-being between intersex-identifying and endosex individuals. Methods: This was a quantitative method pilot study with a cross-sectional approach and exploratory comparative group design. An online survey was used, where a total of 12 self-identifying intersex adult participants were recruited, and 126 endosex adult participants served as a comparative group. Results: The findings show that 83% of the participants reported experiences of discrimination and different types of violence due to their intersexuality. There was a significant difference between the intersex-identifying and endosex groups in psychological well-being, including in three of its dimensions (positives relations, autonomy, and environmental mastery). However, there were no significant differences between the groups in quality of life or social well-being. Conclusion: The findings of this study provide a preliminary understanding of the health disparities of intersex-identifying individuals in Puerto Rico and suggest the need for more profound research, especially the inclusion of other Caribbean and Hispanic countries. The findings also preliminarily imply the need for local and global interventions to reduce physical and mental health disparities and to improve health, quality of life, and well-being among intersex-identifying individuals.

## 1. Introduction

The term intersex is defined as an umbrella term used to describe the differences in the characteristics of physical sexual development [1,2] that do not fit the typical binary notions of mela and female [3]. These characteristics include aspects related to chromosomes, gonads, sex hormones, genitals, internal reproductive organs, and secondary sex characteristics [1]. Approximately between 0.05% and 1.7% of the population are born with intersex traits (also known in medicine as intersex conditions), and 1 in every 2000 neonates presents some degree of genital atypia [2,3]. With the intention of categorizing babies within the sex binary, medically unnecessary surgeries are frequently performed, sometimes even without parental consent [4]. The literature has documented that, occasionally, these surgeries can cause more harm than good [5,6,7,8], and communities and organizations are lining up to fight against non-consensual medical interventions [9]. In other cases, intersexuality is not identified until puberty or adolescence, when primary and/or secondary sex characteristics begin to develop [1].

Scientific research on the intersex spectrum has been increasing in recent years, including that on intersexuality as an identity [10]. However, as our understanding of intersex individuals increases, we are becoming more aware of their most significant health-related issues. One example is maintaining healthy levels of mental health, which, accordingly to the World Health Organization [11], can be experienced as a state of well-being, where an individual realizes their own abilities, can cope with the normal stresses of life, can work productively, and is able to contribute to their community. This state of well-being, more than the absence of mental disorders or disabilities, encompasses the capacity of having joy in life, in interpersonal relationships, and in intrapersonal experiences [11]. Research has shown that individuals with intersex traits have poorer levels of mental health and greater psychiatric diagnoses and distress than endosex individuals (persons with the socially expected characteristics of physical sexual development) [12]. A national study with intersex-identifying individuals in the United States of America [13] showed that over 43% rated their physical health and that 53% rated their mental health as fair or poor. In addition, 28.6% had planned their suicide but did not attempt it; 10.1% attempted it but did not want to die; and 21.7% attempted it and really hoped to die.

The most common clinical diagnoses and symptoms reported in intersex individuals are depression, anxiety [12,13,14,15,16,17,18,19], isolation [14], stress [2], a low self-esteem, a low self-concept, a low quality of life [20], low sexual satisfaction and sexual traumas [2], a low sexual quality of life [21], difficulties in sexual development adaptation [22], difficulties in searching for partners, a lack of identification with a community group, and the feeling of a lack of understanding [23]. Moreover, the suicide attempt rate is 6.8% higher among intersex individuals than endosex individuals [24].

Quality of life is defined as an “individual’s perception of their position in life in the context of the culture and value systems in which they live and in relation to their goals, expectations, standards, and concerns” [25]. It is a multidimensional concept that assesses different aspects of life. Quality of life is divided into many domains, including work, living space, education, neighborhood, community, culture, values, spirituality, and general health [26]. Research in Germany has shown that adult intersex individuals with genital reconstruction normally have a low quality of life in regard to their sexual and relational aspects compared with a sample of women who have not undergone genital reconstruction [21]. Quality of life has also been measured in children and adolescents since it is at this age when changes in growth and puberty affect most intersex individuals [27]. In Brazil, Gilban et al. [27] found a loss of health-related quality of life in intersex children. The self-report was concordant in key areas with their parents’ assessment, which shows coherence between the parents’ and children’s reports.

However, well-being is interpreted as a positive outcome, where people’s perception states that their lives are going well. Thus, psychological well-being is a multidimensional model that focuses on personal development, the style and way of facing life challenges, and the effort and eagerness to achieve goals. It is composed of six dimensions: self-acceptance, positive relationships with other people, autonomy, mastery of the environment, purpose in life, and personal growth [28]. Additionally, social well-being is an individual’s assessment of their circumstances and functioning within society. It is composed of five social dimensions: integration, acceptance, contribution, actualization, and coherence [29].

There is a lack of research about intersex-identifying persons in America, including in Puerto Rico [30,31]. Only a few countries, such as Argentina and Chile, have legislated to protect the rights of intersex individuals and help them receive appropriate care. Colombia, for example, has legislated that cosmetic surgery will only be performed with parental authorization after extensive education related to the process [32]. There are gaps in the research regarding experiences of discrimination and violence, as well as regarding quality of life and psychological and social well-being, among Hispanic intersex individuals, including among those in Puerto Rico.

The aims of this study were (1) to document experiences of discrimination and violence among self-identifying intersex individuals in Puerto Rico and (2) to determine if there is a significant difference in quality of life, psychological well-being, and social well-being between intersex and endosex individuals. Considering prior research on intersex individuals, albeit a small sample, we hypothesized that experiences of violence and discrimination and a lower quality of life and well-being will be reported, similar to global findings.

## 2. Methods

This pilot study used a quantitative method with a cross-sectional approach and a comparison group design. This study was an online survey and used convenience and snowball sampling. Due to the lack of limited resources and knowledge about the term “intersex,” the participation of self-identifying intersex individuals was lower than expected. After a year of recruiting, the team decided to include a community–academic partnership (CAP) to give feedback on the method and design and to support the recruitment process. This approach helped the team to recruit seven new participants.

The selection of participants was based on the following inclusion criteria: (1) being over 21 years old, (2) residing in Puerto Rico, (3) identifying as intersex, (4) knowing how to read and write, and (5) being able to consent to participate. The selection of endosex participants was based on the following inclusion criteria: (1) being over 21 years old, (2) residing in Puerto Rico, (3) knowing how to read and write, and (4) being able to consent to participate; the exclusion criterion was identifying as intersex.

## 3. Instruments

### 3.1. Demographic Data Questionnaire

A 19-item questionnaire was created to collect demographic information, such as age, sex assigned at birth (e.g., male, female, intersex, and other), current gender identity (e.g., male, female, intersex, and other), sex/gender that was assigned on the birth certificate (e.g., male, female, and other), gender expression (e.g., masculinity, femineity, trans, non-binary, and other), sexual orientation, sexual behavior, relationship, place and area of residence, income, functional diversity, religious affiliation, intersex traits, Differences of Sexual Development diagnosis, and chromosomic syndromes. In addition, experiences of discrimination (yes or no) and violence (being harassed on the streets, being sexually abused, being beaten, being followed when walking, and being robbed) were explored. The comparative group’s (endosex persons) demographic data only include the first 13 questions.

### 3.2. World Health Organization Quality of Life Brief Questionnaire (WHOQOL)

The WHOQOL is an instrument developed by the World Health Organization in 1996. This scale measures quality of life in a global way. The WHO initially developed the complete model in English, consisting of 100 items. With the help of 15 collaborating centers in strategic points around the world, the questionnaire was translated and validated in 19 languages and versions, including the WHOQOL-BREF Spanish version. This version consists of 26 items and measures the following scales: physical health (*α* = 0.80), psychological health (*α* = 0.78), social relations (*α* = 0.75), and environment (*α* = 0.78). This instrument was validated with a Spanish population, presenting an acceptable Cronbach alpha for its dimensions (*α* = 0.75–0.080) [24]. The measures are carried out using a five-category response format from 1 to 5 with different options.

### 3.3. Psychological Well-Being Scale (PWBS)

Developed by Ryff, the PWBS is an instrument that measures six dimensions related to psychological well-being. These dimensions include the following areas: self-perception (*α* = 0.83), positive relationships (*α* = 0.81), autonomy (*α* = 0.73), mastery of the environment (*α* = 0.71), purpose in life (*α* = 0.83), and personal growth (*α* = 0.68). To validate this instrument, three researchers generated 80 items per dimension. A total of 32 of the items were administered to 321 adults, and then the items were reduced to 20. Finally, to make the instrument more viable, nine reagents per scale were chosen. The measures are carried out using a six-category response format, from 1 = strongly disagree to 6 = strongly agree. To validate the Spanish version, a sample of 467 people between 18 and 72 years of age was chosen. These participants responded to the PWBS translated in Spanish by Díaz et al. [28]. The scale showed good internal consistency, with Cronbach values between 0.84 and 0.70.

### 3.4. The Social Well-Being Scale (SWBS)

The SWBS was created by Keyes and translated and adapted to Spanish by Blanco and Díaz [29], and it has an internal consistency ranging from α = 0.68 to α = 0.89. This scale consists of 25 items focused on measuring aspects of social integration (α = 0.69), social acceptance (α = 0.89), social contribution (α = 0.70), social updating (α = 0.79), and social coherence (α = 0.68). The measures are carried out using a five-category response format, from 1 = strongly disagree to 5 = strongly agree. The validation was carried out with a sample of 469 participants from the *Universidad Autónoma de Madrid* and 277 workers from different companies in the community of Madrid. The original English scale consisted of 33 items translated into Spanish, where 8 items were eliminated due to low correlation (>0.30) for a final scale of 25 items.

## 4. Recruitment and Procedure

Upon Institutional Review Board approval (#1903007095) at Ponce Health Sciences University, we used a variety of approaches to recruit self-identifying intersex participants from around the island. We used Facebook ads to promote the study as a convenience sampling method and delivered flyers to cooperating endocrinologists, gynecologists, urologists, and geneticists in the three main cities of the island (San Juan, Ponce, and Mayagüez). Given that the LGBT+ and sexually diverse communities in Puerto Rico are interconnected, snowball sampling was likely to occur by having members of the trans community promote the study. Later on, our community member created educational videos and infographics about intersexuality to promote the project flyer.

To recruit the comparative group (endosex persons), we used a separate Facebook ad based on a convenience sampling method. The REDCap platform was used to access the anonymous survey. After accepting the platform’s informed consent, the participants were able to complete the sociodemographic questionnaire and instruments.

## 5. Statistical Analyses

Our study population was described by group using relative and absolute frequencies. The distribution of the data was assessed using the Shapiro–Wilk test and Bartlett’s test prior to any statistical analyses. Nonetheless, non-parametric alternatives were chosen due to the small sample size of the intersex group and because not all variables fulfilled the assumptions for normality. Associations for categorical variables were assessed using Fisher’s Exact Test, while continuous variables were assessed using the Mann–Whitney Test. *p*-values ≤ 0.05 were considered statistically significant. All statistical analyses were carried out using STATA SE 16.

## 6. Results

### 6.1. Participants

#### 6.1.1. Intersex Group

A total of 13 self-identifying intersex participants contributed to this study; however, only 12 of them completed the instruments (92%). The mean age of this group was 34 years (*SD* = 14.65), with a range of 21 to 69. In addition, the majority of the group was assigned female at birth (58.3%; *f* = 7), currently identify their gender as female (50%; *f* = 6), and identify their gender expression as feminine (45.5%; *f* = 5). While documenting sex trait variations or characteristics to identify as intersex, 91.7% (*f* = 11) indicated hormonal characteristics, 50% (*f* = 6) genitalia, 41.7% (*f* = 5) internal reproductive organs, 33.3% (*f* = 4) chromosomal characteristics, and 16.7% (*f* = 2) gonadal characteristics. Only one participant (8.3%) reported having a Differences of Sexual Development (DSD) diagnosis. Regarding their sexual orientation, 25% (*f* = 3) identify as bisexual, 25% (*f* = 3) as gay or lesbian, 16.7% (*f* = 2) as heterosexual, 16.7% (*f* = 2) as pansexual, and 16.7% (*f* = 2) as other (demisexual and “pansexual bisexual and asexual”). When exploring sexual behavior, it was found that 16.7% (*f* = 2) have never had a sexual experience, 83.3% (*f* = 10) have had sex with men, 58.3% (*f* = 7) have had sex with women, and 16.7% (*f* = 2) have had sexual relations with another intersex person.

Moreover, 41.7% (*f* = 5) reported having a partner, 91.7% (*f* = 11) were from an urban area, 25% (*f* = 3) disclosed a functional diversity, and 66.7% (*f* = 8) had a religion or spiritual affiliation. Areas of residence were also reported, with 58.3% (*f* = 7) living in the metropolitan area. Moreover, 91.7% (*f* = 3) reported an approximate income below USD 30,000 (see Table 1).

#### 6.1.2. Endosex Group

For the comparative group, 126 participants completed the questionnaire. The average age of this group was 36 years (*SD* = 12.59), with a range of 21 to 71. In addition, the majority of the group was assigned female at birth (86.5.3%; *f* = 109) and currently identify their gender as female (84.9%; *f* = 107) and their gender expression as feminine (82.5%; *f* = 104). Regarding their sexual orientation, 78.6% (*f* = 99) identify as heterosexual, 10.3% (*f* = 13) as bisexual, 7.9% (*f* = 10) as gay or lesbian, 2.4% (*f* = 1) as pansexual, and 0.8% (*f* = 1) as asexual. When exploring sexual behavior, it was found that 0.8% have never had a sexual experience, 90.5% (*f* = 114) have had sex with men, 21.4% (*f* = 27) have had sex with women, and 0.8% have had sex with an intersex person.

In addition, 62.7% (*f* = 79) reported to have at least one partner, 49.2% (*f* = 62) were from an urban area, 9.5% (*f* = 12) disclosed a functional diversity, and 55.6.3% (*f* = 70) had a religion or spiritual affiliation. Areas of residence were also reported, with 18% (*f* = 23) living in the metropolitan area. Moreover, 84.1% (*f* = 106) reported an approximate income below USD 30,000 (see Table 1).

### 6.2. Internal Consistency of the Instruments

The analyses revealed that all scales obtained adequate values of internal consistency higher than 0.90 (WHOQOL, *α* = 0.91, PWBS, *α* = 0.94, and SWBS, *α* = 0.94). A series of Fisher Exact Tests were conducted to examine the comparability of the groups (see Table 1).

### 6.3. Aim 1

As expected, 83.3% of the participants reported experiences of discrimination related to their intersex identity. Experiences of violence due to their intersexuality were also reported in all options: being harassed on the streets (50%), being sexually abused (50%), being followed when walking (50%), being beaten (33.3%), and being robbed (33.3%).

### 6.4. Aim 2

#### 6.4.1. Quality of Life

There was no significant difference in the levels of quality of life between the intersex (*Mdn* = 72.5) and endosex (comparative) (*Mdn* = 81) groups (*U*(*N*_inter_ = 12, *N*_endo_ = 126) = 591.50, z = −1.243, *p* = 0.0214). There were also no significant differences in the quality of life dimensions of social relations (*U*(*N*_inter_ = 12, *N*_endo_ = 126) = 794.00, z = 0.288, *p* = 0.773); physical health (*U*(*N*_inter_ = 12, *N*_endo_ = 126) = 606.00, z = −1.137, *p* = 0.255); psychological health (*U*(*N*_inter_ = 12, *N*_endo_ = 126) = 582.50, z = −1.317, *p* = 0.0188); and environmental (*U*(*N*_inter_ = 12, *N*_endo_ = 126) = 593.50, z = −1.230, *p* = 0.219) (see Table 2 and Appendix A).

#### 6.4.2. Psychological Well-Being

There was a significant difference in the levels of psychological well-being between the intersex (*Mdn* = 46.13) and endosex (comparative) (*Mdn* = 71.73) groups (*U*(*N*_inter_ = 12, *N*_endo_ = 126) = 475.50, z = −2.120, *p* = 0.034). There was also a significant difference in the psychological well-being dimensions of positive relations (*U*(*N*_inter_ = 12, *N*_endo_ = 126) = 487.00, z = −2.036, *p* = 0.042); autonomy (*U*(*N*_inter_ = 12, *N*_endo_ = 126) = 490.50, z = −2.009, *p* = 0.045); and environmental mastery (*U*(*N*_inter_ = 12, *N*_endo_ = 126) = 394.00, z = −2.740, *p* = 0.006). However, the self-acceptance (*U*(*N*_inter_ = 12, *N*_endo_ = 126) = 531.00, z = −1.703, *p* = 0.089); purpose in life (*U*(*N*_inter_ = 12, *N*_endo_ = 126) = 655.00, z = −764, *p* = 0.445); and personal growth (*U*(*N*_inter_ = 12, *N*_endo_ = 126) = 785.50, z = 0.225, *p* = 0.822) dimensions were not significant (see Table 2 and Appendix A).

#### 6.4.3. Social Well-Being

There was no significant difference in the scores for social well-being between the intersex (*Mdn* = 46.05) and endosex (comparative) (*Mdn* = 70.28) groups (*U*(*N*_inter_ = 12, *N*_endo_ = 126) = 405.50, z = −1.872, *p* = 0.061). Despite this result, there was a significant difference in the social well-being dimensions of social integration (*U*(*N*_inter_ = 12, *N*_endo_ = 126) = 360.50, z = −2.253, *p* = 0.024). However, the social contribution (*U*(*N*_inter_ = 12, *N*_endo_ = 126) = 449.50, z = −1.513, *p* = 0.130); social actualization (*U*(*N*_inter_ = 12, *N*_endo_ = 126) = 523.50, z = −0.890, *p* = 0.374); social acceptance (*U*(*N*_inter_ = 12, *N*_endo_ = 126) = 413.50, z = −1.812, *p* = 0.070); and social coherence dimensions (*U*(*N*_inter_ = 12, *N*_endo_ = 126) = 636.50, z = 0.054, *p* = 0.957) were not significant (see Table 2 and Appendix A).

## 7. Discussion

This pilot study is one of the first steps to generate preliminary data and knowledge about self-identifying intersex individuals’ health disparities in Puerto Rico. It should be noted that the intersex-identifying participants seemed to have participated because they self-identify as intersex and not because they have been diagnosed with DSD. The team had limitations recruiting the minimum number of participants, which could mean that it is not feasible to recruit intersex-identifying persons in Puerto Rico for larger or generalizable quantitative studies. However, the sample could be suitable for qualitative studies. Overall, our study found that the intersex-identifying participants reported more experiences of violence and discrimination and lower levels of quality of life, psychological well-being, and social well-being than the comparative group of endosex participants, but only the differences in psychological well-being were statistically significant. These results are not consistent with various studies, where individuals with intersex traits reported a significantly lower quality of life [21,25,27]. Therefore, it is congruent with other research that presents a significantly reduced well-being [33]. We hypothesize that these results could be explained, as most studies have largely explored the association between a lower quality of life and well-being in individuals with a DSD diagnosis and/or who have undergone genital surgeries related to intersexuality [7,8]. However, our study focused on the psychological dimension of identity (self-identifying intersex individuals), and information on the presence of any surgery was not collected.

Quality of life is part of an individual’s health. The intersex-identifying participants reported a lower quality of life, including a lower quality of life in its dimensions; however, those differences were not statistically significant. Studies that report significant differences in quality of life among individuals with intersex traits tend to associate these with difficulties in finding a partner, a lack of identification with a community group, and the feeling of a lack of understanding [22]. Almost half of our sample reported having a partner, and the majority also identified as a sexual minority (e.g., lesbian, gay, bisexual, and pansexual). This characteristic may have influenced the results.

In addition, the literature supports the notion that, although participants usually report good physical health, they experience more general health problems [13]. On the contrary, individuals with intersex traits in other studies have exhibited diminished sexual responsiveness and pleasure due to surgical procedures and may have discomfort associated with repeated surgical procedures and humiliation by physicians [34]. This is why it is important to explore these surgical procedures and the barriers that they create.

Regarding psychological well-being, significant differences were found, including in the dimensions of positive relations, autonomy, and environmental mastery. Bohet et al. [12] showed that individuals with intersex traits report worse mental health levels than endosex individuals. Research has also documented that individuals with intersex traits report low self-esteem [20] and anxiety due to their physical appearance [16]. Since the environmental dimension measures an individual’s ability to manage their life and surroundings [35], individuals with intersex traits could feel a lack of control in life by living in a binary society and a culture that makes decisions on other people’s bodies, sometimes without consent. The dimension of autonomy can also be explained by the previous reason, as intersex individuals can lose their feeling of independence.

Moreover, social well-being showed no significant differences. As we discussed earlier, our sample was mostly composed of people with a partner. In addition, the participants self-identified as intersex; therefore, assuming and living this identity could provide benefits in terms of acceptance, contribution, and social coherence, increasing their social well-being. Previous research, including Schönbucher et al. [21], found that individuals with intersex traits may have difficulties finding a partner and are considered more insecure in social and sexual situations. Likewise, Ediati et al. [15] found that women with intersex traits tend to isolate themselves socially. This could explain why our sample reported significantly lower levels of social integration.

### 7.1. Strengths and Limitations

Our pilot study has strengths and limitations that deserve to be acknowledged. Some of the strengths are as follows: the intersex-identifying group was diverse in terms of sex and gender identity, the study had a comparative group of endosex individuals that allowed us to better describe the variables, and the internal consistency of the instruments was adequate. However, the limitations are as follows: it was a small non-probabilistic sample due to the difficulty of recruiting intersex-identifying individuals in Puerto Rico. Recruiting intersex-identifying individuals for the study was extremely challenging. We believe that there is a wide-ranging lack of knowledge and misinformation about the term in the general population. Thus, the results do not represent the population of people in Puerto Rico with intersex traits or those who identify as intersex in the absence of evidence of DSD. Only one participant reported a DSD diagnosis; therefore, comparisons between the participants with and without the diagnosis were not possible. In addition, minors under 21 years of age were not included, and those who identify as male, with a masculine identity, and as a sexual minority (LGB+) were significantly underrepresented in the endosex group when compared with the intersex group.

### 7.2. Future Directions

For futures studies, we recommend expanding the sample using community outreach and exploring physical and mental health disparities and the social determinants of health impacting the health, quality of life, and well-being of intersex-identifying individuals and individuals with intersex traits, as well as individuals with DSD diagnoses for comparison. Researching surgeries related to intersexuality, barriers to health care, and perceived stigma are also recommended. Qualitative in-person interview methods are also encouraged to better understand what is categorically impacting their quality of life, well-being, and physical and mental health from their own perspective. Longitudinal studies could also be useful to measure outcome changes over time. Studies using medical record searches for DSD diagnoses in databases are also encouraged. It is essential to continue researching this vulnerable population, specifically the protective factors of their overall health.

### 7.3. Conclusions

This is the first study in Puerto Rico and one of the first in Latin America that measures quality of life and well-being among intersex-identifying adult individuals. Due to the nature of an experimental study, we cannot make conclusions or generalizations about the results, but we understand that the findings of this study enrich preliminary awareness/knowledge of health disparities among intersex-identifying individuals on the island. Our findings also suggest the need for more research, especially in other Hispanic countries and Hispanic individuals living in the continental United States of America. The findings also imply the need for local and global interventions to reduce physical and mental health disparities and to improve health, quality of life, and well-being among intersex individuals.

## Figures and Tables

**Table 1 ijerph-20-02899-t001:** Demographic Characteristics of the Participants.

Variables	Total n= 138 *f* (%)	Intersex		*p* ¹
		No *n* = 126 *f* (%)	Yes *n* = 12 *f* (%)	
Place of Birth				0.005 *
Urban	73 (52.9)	62 (49.2)	11 (91.7)	
Rural	65 (47.1)	64 (50.8)	1 (8.3)	
Sex Assigned at Birth				0.087
Male	21 (15.2)	17 (13.5)	4 (33.3)	
Female	117 (84.8)	109 (86.5)	8 (66.7)	
Current Gender Identity				0.002 *
Male	22 (15.9)	17 (13.5)	5 (41.7)	
Female	115 (83.3)	109 (86.5)	6 (50.0)	
Intersex	1 (0.7)	-	1 (8.3)	
Gender Expression				0.010 *
Masculine	22 (16.06)	18 (14.3)	4 (36.4)	
Feminine	109 (79.56)	104 (82.5)	5 (45.5)	
Transgender or Non- Binary	6 (4.38)	4 (3.2)	2 (18.2)	
Sexual Orientation				<0.001 *
Heterosexual	101 (74.3)	99 (78.6)	2 (20.0)	
LGB+	35 (25.7)	27 (21.4)	8 (80.0)	
Income				0.948
<USD 10,000	61 (44.2)	56 (44.4)	5 (41.7)	
USD 10,000–USD 30,000	56 (40.6)	50 (39.7)	6 (50.0)	
USD 30,000–USD 50,000	15 (10.9)	14 (11.1)	1 (8.3)	
USD 50,000 or more	6 (4.4)	6 (4.8)	0	
Functional Diversity (Impairment)				0.125
No	123 (89.1)	114 (90.5)	9 (75.0)	
Yes	15 (10.9)	2 (40.0)	3 (25.0)	
Religious Affiliation				0.552
No	60 (43.5)	56 (44.4)	4 (33.3)	
Yes	78 (56.5)	70 (55.6)	8 (66.7)	

Note: ¹ *p*-values were obtained using Fisher Exact Test. * Statistically significant values (*p* < 0.05).

**Table 2 ijerph-20-02899-t002:** Independent Samples’ Mann–Whitney Test Results Comparing Quality of Life and Psychosocial Well-Being Between Intersex and Endosex Groups.

Variables	Total *n = 138* Median (p25–p75)	Groups		*p* ¹
		Endosex *n* = 126 Median (p25–p75)	Intersex *n* = 12 Median (p25–p75)	
Quality of Life	81 (69–93)	81 (69–93)	72.5 (62–91)	0.214
Physical Health	21 (18–24)	21 (18–24)	20.5 (19–24)	0.773
Psychological Health	19 (17–22)	20 (17–22)	17.5 (16–21.5)	0.255
Social Relations	9 (7–11)	9 (7–11)	7.5 (6–9.5)	0.188
Environmental	25 (21–30)	26 (21–30)	23.5 (17.5–29)	0.219
Psychological Well-Being	126.5 (104–146)	129 (105–148)	106 (83.3–125.5)	0.034 *
Self-Acceptance	21 (14–26)	22 (15–26)	17 (11.5–21.5)	0.089
Positive Relations	18 (14–24)	18 (15–24)	14.5 (9.5–19)	0.042 *
Autonomy	26 (20–31)	27 (21–31)	21.3 (17.5–25.5)	0.045 *
Environmental Mastery	20.5 (16–24)	21 (16–25)	15 (13–19)	0.006 *
Purpose in Life	23 (15–26)	23 (15–26)	20.5 (9–26)	0.445
Personal Growth	19 (16–23)	19 (16–23)	20.5 (15.5–22.5)	0.822
Social Well-Being	86 (70.5–96)	87 (73–98)	74 (54–89)	0.061
Social Integration	18 (14.5–21)	18 (15–21)	12 (9–17)	0.024 *
Social Acceptance	16 (12–20)	16.5 (12–20)	15 (8–18)	0.374
Social Contribution	20 (16–24)	20.5 (16–24)	15 (9–24)	0.130
Social Actualization	16 (14–20)	16.5 (14–20)	13.5 (12–15)	0.070
Social Coherence	15 (12.5–18)	15 (13–18)	15 (12–19)	0.957

Note: ¹ *p*-values using Mann–Whitney Test. * Statistically significant values (*p* < 0.05).

## Data Availability

Please contact PI: cesteban@psm.edu.

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
