# Peer review of "Quality of Life and Psychosocial Well-Being among Intersex-Identifying Individuals in Puerto Rico: An Exploratory Study"

_ijerph, 2023, doi:10.3390/ijerph20042899_

Round 1

Reviewer 1 Report

I praise the paper's authors for the timeliness and importance of the work developed. In an effort to try and add something of value to that work, I present below a few details that might require some further editing.

Spell checking

There are a few spelling errors of note in the manuscript, namely,

line 119 - identify

line 122 - collet

line 152 - improper citation style

line 156 - improper citation style

line194 - bird

line 194 - two percentages for seemingly the same thing (50% and 20%)

line 198 - Either Difference or Disorder, although Difference seems to be more in line with the paper

line 218 - man

line 321 - missing a "of"

line 328 - an extra "others"

Some clarity issues

Section 7, especially lines 277-283 are a bit confusing to read, especially when comparing this study to different studies. It would be interesting to hypothesize why there are / aren't differences.

While explaining different results, the paper resorts to the same explanation (lines 286-288 vs lines 305-307).

This section does not adequately discuss the potential reasons for why the results are as they are.

Reviewer 2 Report

This study addresses an important issue; however, it has many shortcomings.

 The title is “Quality of live and psychosocial well-being among intersex individuals in Puerto Rico: an exploratory study. “ 

 Major issues

The major limitation of the study is that it employed a small sample of individuals (n=13) who identified as intersex, only one of whom reported having been diagnosed with a difference/disorder of sex development (DSD)).  In this reviewer’s experience, few people who identify as intersex actually have a DSD/intersex condition.  As a clinician with experience working in this area, the majority of patients who were referred to me for evaluation after having identified to another clinician as being intersex, did not have a verifiable DSD, but most continued to identify as intersex/having a DSD after the evaluation.  I quickly learned not to try to convince such individuals that they did not have a DSD but to assure them that they did not have a condition that was in need of treatment without challenging their identity as intersex.  Also, many DSDs do not result in ambiguity regarding sexual differentiation or sex assignment so that the terms “intersex condition/disorder” and “difference/disorder of sex development” are not equivalent.  For example, mild hypospadias is a DSD but not an intersex condition.  In the absence of clinical confirmation of an intersex DSD, I do not have much confidence in the data.  Until fairly recently in the US it was not that unusual for individuals to state they had an intersex DSD as they believed it would facilitate access to gender affirming surgeries.

 Moreover, historically, the goals of the intersex movement - minimizing genital surgeries, were sometimes at odds with the goals of the transgender movement – preventing unnecessary surgeries.  Intersex movements tend to emphasize the limitations of surgery together with autonomy, while transgender movements tend to work toward increased access to surgery.  As a whole, intersex goals and transgender goals have major differences, with some areas of overlap.

 If this submission is to be considered for publication the above limitations should be addressed in light of of the recruitment methods, along with implications for future research.  

The frequency of particular intersex conditions in PR might be queried in future research by examining diagnostic codes in the medical records of larger hospitals and medical clinics.  That would not be a simple task, given the large number of different DSDs; however, perhaps codes for the more frequent conditions could be queried.  Intersex/DSD advocacy organizations may be of assistance in identifying intersex individuals and others with DSD who live in PR or grew up there.

                                               Other Issues

Terminology

People should not be described as intersexual or intersex but has having an intersex condition, when such a condition has been verified. In the absence of verification, subjects should be described as reporting that they are intersex or identifying as intersex.   Since no verification of a DSD diagnosis was available for the entire sample, the sample might best be described as “identifying as intersex.”  Note, that many people with a DSD do not identify as intersex but as a person with a DSD.

Similarly, “intersex persons” should be people with intersex conditions, or given the present sample, perhaps “intersex-identified persons.”

 When addressing the need to study the health disparities of individuals with intersex conditions, you might state “physical and mental health disparities.”

 “Sexual” refers to sexuality.  It would be preferable to refer primary and/or secondary sex characteristics rather than “sexual characteristics.”

 Also avoid the term intersexuality.  Intersex conditions would be better.

Introduction

The first sentence of the second paragraph of the Introduction is ambiguous.

The discussion of surgical outcomes is limited.  It is not helpful unless particular surgeries on people with particular conditions are discussed.

Sixth sentence of Intro:  Recommend omitting “Unfortunately” and replacing “adequate’ with “appropriate.”

 The hypothesis should be stated differently in the last sentence of the Intro. This sentence does not state a hypothesis as written.  A hypothesis is something to be tested.  It is not necessary that you test a hypothesis, however.  Perhaps, you could just state the purpose – to examine experiences of …….  You might then comment that based on prior research on individuals with intersex conditions and/or who identified as intersex, you hypothesized that ….,.

Instruments

Replace “actual sex identity” with “current gender identity.”

Recruitment Procedure

Many people with intersex DSDs (depending on the particular diagnosis) do not identify as members of the sexual and gender minority communities but as cisgender/heterosexual people with a medical condition (e.g., many people with complete androgen insensitivity syndrome).

In my experience, many in the transgender community identify as intersex because they believe they have intersex brains, something that cannot be objectively demonstrated.  Most in my experience, based their belief on their history of childhood gender nonconformity and a desire to be the other gender.  This is a variation of the born that way hypothesis – one believes that they must have a brain that is of a different sex compared with their genitalia.

Results

As genetic/chromosomal variation can produce hormonal variations that impact development of the external genitalia, it might be interesting to add a chart indicating how many of these factors each individual endorsed – perhaps as a supplementary table.  The table could also include sexual and gender identity as well as sexual behavior.

Discrimination due to intersexuality lacks specificity, and if no more specificity can be stated, this should be addressed in the limitations.  Specifically, I would like to know if the discrimination, etc. resulted from gender atypical appearance (when clothed), gender atypical mannerisms, atypical genitalia in the locker room or when playing doctor, etc.

The absence of differences in quality of life is surprising – especially since there was a difference in psychological well-being.  This should be addressed.

Discussion

The fact that only one participant identified as having been diagnosed as having a DSD does not surprise me.  This has to be appropriately addressed in the Limitations. 

It is not clear what “representative in sex and gender identity” means.  Representative of what?  Do you mean varied in this regard. 

Throughout, you should indicate whether you are interested in people who identify as intersex or people with intersex DSDs.  If you are interested in the former, you must indicate why.   Going forward, you might want to consider focusing on people who identify as intersex or as having a DSD to get larger numbers and then stratify by those with and without a verified DSD diagnosis. 

Medical record searches for DSD diagnoses in large databases followed by chart review, and in person interviews, when possible, would be most helpful.

Round 2

Reviewer 2 Report

I was surprised to receive this revised report so soon after my review. The revision appears to have been prepared hastily. There are multiple minor issues with English usage scattered throughout the report. I assume these can be corrected by copy editors. Some (e.g., singular vs plural) are of the nature that would be picked up by using the Editor function in Word. The authors are encouraged to use Editor prior to resubmission. More problematic, there still remain passages that do not make it clear that the participants in this study identified as intersex. Only one subject  could name their intersex condition, but that was not verified by checking their medical record. The authors must not suggest or appear to assume at any point in the manuscript that their subjects actually had intersex conditions, aka differences or disorders of sex development.  They identified as intersex.

The text of the online survey should be included, perhaps as supplementary material.

 Particular points that must be revised include:

 1)      In the Abstract and throughout, do not refer to the subjects as intersexual individuals. They can be described as intersex-identified individuals.

 2)      The Conclusions of the Abstract refers to “intersex individuals” TWICE. At no place in the manuscript should participants be referred to as intersex. At best the population should be referred to as “intersex-identified” individuals. It is worth noting that as a clinician my experience has been that few patients with intersex diagnoses identify as intersex. Instead, most identify as a man or woman/boy or girl according to the sex they were assigned at birth and as a person with a medical condition – which they can describe if they cannot name. Most adults, I have seen who believed they had an intersex condition did not have any evidence of such.

The text revision of the DSM-5 (DSM-5TR) now requires that the specifier "with a disorder/difference of sex development" be used for individuals with gender dysphoria ONLY when a specific DSD can be coded. 

 3)      The first line of the introduction must define “intersex” appropriately. It is currently defined as “…a term to describe the diversity or differences in characteristics of the physical sexual development.”

“Intersex” does not describe diversity. Intersex reflect the diversity of sex development. The term, intersex conditions, should be defined as a set of somatic conditions that involve variations in the development of the sexual characteristics of the body which may include chromosomes, gonads and genitalia that do not fit typical binary notions of male and female bodies [Free & Equal Campaign Fact Sheet: Intersex" (PDF). United Nations Office of the High Commissioner for Human Rights. 2015.]

 The authors are encouraged to read the appendix on intersex conditions that was published by a taskforce report by of the American Psychiatric Association which is freely available online at Transgend Health. 2018; 3(1): 57–A3. Published online 2018 May 18. doi: 0.1089/trgh.2017.0053. This report addresses the fact that some people identify as intersex in the absence of any evidence of an intersex condition.  People are entitled to identify any way they chose; however, it is important to note that “intersex” has become an identity label.  Not all with intersex DSDs identify as intersex, and many who do identify as intersex do not have a DSD.

 In this reviewers’ experience, adolescents and adults in the US who have an intersex condition, not just an intersex identity, can usually name it.

4)      Line 94. “management” should be changed to “care.”

5)      Under Instruments, line 125-126. It is more customary to query sex/gender assigned at birth, and current gender identity. Depending on the jurisdiction, birth certificates have historically usually indicated either sex or gender but not both. Medicolegally, they have been considered equivalent.

 6)      Discussion. Line 285 and elsewhere as relevant. Again, it is probably still true that the majority of individuals with intersex conditions do not identify as intersex, but as a man or woman who can name their condition. Therefore, rather than “intersex individuals”, the where appropriate the text should read ”Individuals with intersex conditions”

7)      L 305 “differences were significantly found” should be “significant differences were found.”

 8)      331-32 “Results do not represent the intersex population” should be “Results do not represent either the population of people in Puerto Rico with intersex conditions or those who identify as intersex in absence of evidence of an intersex condition.

 9)      Discussion l 347. “Vulnerablized” is not a word. Replace with “vulnerable.”

10)      You might comment that such studies could identify similarities between individuals with intersex differences of sex development (DSD) and those who identify as intersex in the absence of an intersex DSD. Note that all DSDs do not call into question which sex should be assigned at birth so that while all intersex conditions are DSDs, not all DSDs are intersex conditions.
